# Definition of Biologically Distinct Groups of Conjunctival Melanomas According to Etiological Factors and Implications for Precision Medicine

**DOI:** 10.3390/cancers13153836

**Published:** 2021-07-30

**Authors:** Sophie Gardrat, Alexandre Houy, Kelly Brooks, Nathalie Cassoux, Raymond Barnhill, Stéphane Dayot, Ivan Bièche, Virginie Raynal, Sylvain Baulande, Richard Marais, Sergio Roman-Roman, Marc-Henri Stern, Manuel Rodrigues

**Affiliations:** 1INSERM U830, DNA Repair and Uveal Melanoma (D.R.U.M.), Equipe Labellisée par la Ligue Nationale Contre le Cancer and PSL Research University, Department of Biopathology, Institut Curie, PSL Research University, F-75005 Paris, France; sophie.gardrat@curie.fr; 2INSERM U830, DNA Repair and Uveal Melanoma (D.R.U.M.), Equipe Labellisée par la Ligue Nationale Contre le Cancer, Department of Genetics, Institut Curie, PSL Research University, F-75005 Paris, France; alexandre.houy@curie.fr (A.H.); stephane.dayot@curie.fr (S.D.); marc-henri.stern@curie.fr (M.-H.S.); 3Molecular Oncology Group, CRUK Manchester Institute, The University of Manchester, Alderley Park, Manchester M13 9PL, UK; Kelly.Brooks@qimrberghofer.edu.au (K.B.); Richard.Marais@cruk.manchester.ac.uk (R.M.); 4QIMR Berghofer Medical Research Institute, Brisbane, QLD 4006, Australia; 5Department of Ocular Oncology, Faculty of Medicine, Institut Curie, Université de Paris Descartes, F-75005 Paris, France; nathalie.cassoux@curie.fr; 6Department of Biopathology, Institut Curie, PSL Research University, F-75005 Paris, France; raymond.barnhill@curie.fr; 7INSERM U1016, Institut Curie, Department of Genetics, Faculty of Pharmaceutical and Biological Sciences, Université de Paris, F-75005 Paris, France; ivan.bieche@curie.fr; 8Institut Curie Genomics of Excellence (ICGex) Platform, Institut Curie, PSL Research University, F-75005 Paris, France; virginie.raynal@curie.fr (V.R.); sylvain.baulande@curie.fr (S.B.); 9Translational Research Department, Institut Curie, PSL Research University, F-75005 Paris, France; sergio.roman-roman@curie.fr; 10Department of Medical Oncology, Institut Curie, PSL Research University, F-75005, Paris, France

**Keywords:** conjunctival melanoma, sun exposure, nevus, primary acquired melanocytosis, *BRAF*, *NRAS*, *KIT*, *CTNNB1*

## Abstract

**Simple Summary:**

Conjunctival melanoma (ConjMel) is a rare but potentially deadly eye tumor developing on the ocular mucosal surface, which is partially exposed to sunlight. The relationships between potential etiological factors such as ultraviolet exposure and ConjMel mutational landscape have not been precisely described in large cohorts. Here, we report the sequencing of 400 cancer genes in 47 primary ConjMel and show several associations between mutations and etiological factors. *BRAF*- and *CDKN2A*-mutated ConjMel affect younger patients while *NF1*-mutated tumors tend to develop in older ones. *CTNNB1* mutations are more frequent in nevi-derived ConjMel, suggesting that the Wnt pathway is pivotal in their tumorigenesis. We further identified concomitant *KIT*/*SF3B1* mutations in *BRAF*/*RAS*-wild type, sun-protected tumors, suggesting a similar profile as previously observed in genital and anorectal melanomas, thus unveiling a distinct, mucosal-specific, tumorigenic pathway. Finally, we report for the first time new targetable oncogenic mutations, opening new therapeutic options for these patients.

**Abstract:**

Conjunctival melanoma (ConjMel) is a potentially deadly ocular melanoma, originating from partially sunlight-exposed mucosa. We explored the mutational landscape of ConjMel and studied the correlation with etiological factors. We collected 47 primary ConjMel samples and performed next-generation sequencing of 400 genes. Hotspot mutations in *BRAF*, *NRAS*, *HRAS*, and *KIT* were observed in 16 (34%), 5 (11%), 2, and 2 cases, respectively. Patients with *BRAF* and *CDKN2A*-mutated ConjMel tended to be younger while the *NF1*-mutated one tended to be older. The eight tumors arising from nevi were enriched in *CTNNB1* mutations (63% vs. 8%; Fisher’s exact *p*-test = 0.001) compared to non-nevi ConjMel and five were devoid of *BRAF*, *RAS*, *NF1*, or *KIT* mutations, suggesting a specific oncogenic process in these tumors. The two *KIT*-mutated cases carried *SF3B1* mutations and were located on sun-protected mucosa, a genotype shared with genital and anorectal mucosal melanomas. Targetable mutations were observed in *ERBB2*, *IDH1*, *MET*, and *MAP2K1* (one occurrence each). Mutational landscape of ConjMel characterizes distinct molecular subtypes with oncogenic drivers common with mucosal and skin melanomas. *CTNNB1* mutations were associated with nevus-derived ConjMel. Concomitant *KIT*/*SF3B1* mutations in sun-protected cases suggest a common tumorigenic process with genital and anorectal mucosal melanomas.

## 1. Introduction

Melanomas are a heterogeneous group of tumors that may arise from diverse tissues including glabrous or non-glabrous skin, mucosae, uvea, or leptomeninges. These melanoma subtypes present distinct clinical behaviors. For instance, cutaneous melanomas (CM) usually spread in lymph nodes before progressing to almost any viscera [1], while uveal melanomas undergo hematogenous spread and exhibit a strong liver tropism [2]. Ultraviolet (UV)-induced damage is another distinguishing factor between melanoma subtypes. CM are ultraviolet-induced tumors, typically exhibiting high mutation burden and proportion of CC > TT transitions, which are absent in most other melanomas [3]. Distinct genetic patterns also define these melanoma subtypes. Epithelia-associated melanomas such as cutaneous, acral, and mucosal melanomas frequently display mutations activating the MAPK, KIT, MITF, and/or TERT pathways, and genetic alterations inactivating tumor suppressor genes such as *CDKN2A*, *NF1*, *TP53*, and/or *PTEN* [3,4]. CM have been classified in four subtypes defined by *BRAF* mutations (~50% of cases), *NRAS/HRAS* mutations (~25%), *NF1* variants (~10%), and the absence of these variants (~15%). Clinical and genetic features further vary between CM subtypes, with CM arising from chronically sun-damaged skin such as facial CM occurring in older patients, which is associated with a higher frequency of *BRAF*^nonV600E^, *NRAS*, *NF1*, and *KIT* mutations [5]. The frequencies of these genetic aberrations also differ depending on the tissue of origin of these melanomas, for instance, *KIT* mutations are more common in mucosal melanomas. Importantly, these mutations are not observed in uveal and leptomeningeal melanomas, which are instead commonly mutated in the G-protein coupled receptor pathway including *GNAQ*, *GNA11*, *CYSLTR2*, and *PLCB4* [6]. Altogether, these genetic alterations influence clinical decisions for metastatic disease management with BRAF and MEK inhibitor treatment being suitable for patients with *BRAF*-mutated melanomas, while patients with *KIT*-mutated melanomas benefit from KIT inhibitors [7]. Furthermore, immune checkpoint inhibitors have shown a high activity in CM because of their high clonal mutation burden, whereas their efficacy is reduced in mucosal melanoma, a subtype carrying a much lower mutation burden [8,9].

Conjunctival melanoma (ConjMel) arises from conjunctiva, the ocular external mucosa, mostly within areas of primary acquired melanocytosis (PAM) (~65% of cases), in conjunctival nevi (~15% of cases) or de novo in ~20% of cases [10,11,12]. ConjMel represents around 5% of ocular melanomas and 0.25% of all melanomas but its incidence has been increasing over the last decades with a suspected relationship to sunlight exposure [13,14]. ConjMel risk factors are close to those of CM including genetic predisposition, potential UV exposure at early ages, and fair skin [12]. Although ConjMel mostly occurs on sunlight-exposed areas, they may also appear on sunlight-protected areas (i.e., behind eyelids) [12,15]. Mitomycin C instillations, interferon, and radiotherapy are often used in the adjuvant setting after tumor excision to reduce the risk of metastases, local recurrence rate, and therefore the risk of exenteration. Even with optimal local treatment, approximately 20–30% of ConjMel patients will eventually develop regional and visceral metastases with a pattern similar to CM [10,16]. Therefore, ConjMel is a potentially sight- and life-threatening disease. While the genetics of other melanomas and especially CM have been much described in recent years, there is a paucity of large cohort genetic ConjMel studies. Recent reports have shown that ConjMel presents an UV-associated mutational signature together with a high number of indels and copy number alterations [17,18,19]. Other reports have described *BRAF* activating mutations in approximately a third of patients, *NRAS* Q61 in 0 to 18%, and *KIT* in less than 5% [20,21,22,23,24,25]. The purpose of this study was to describe the ConjMel mutational landscape in order to identify specific oncogenic mechanisms and druggable targets.

## 2. Materials and Methods

### 2.1. Sample Collection

This study was approved by the Internal Review Board of Institut Curie. Tumors were deemed “non-sunlight-exposed” if the entire lesion was behind the eyelids, and “sunlight-exposed” if at least a part of the lesion was on a sun-exposed area. All analyzed samples came from 47 independent archived formalin-fixed paraffin-embedded (FFPE) samples.

### 2.2. DNA Extraction

Samples were histologically reviewed by a pathologist before nucleic acid extraction in order to select samples with at least 30% of tumor cells. DNA was extracted by the Centre de Ressources Biologiques (Institut Curie, 26 rue d’Ulm, 75248 Paris, France, tumor biobank) from FFPE using the Nucleospin Tissue Kit (Macherey-Nagel GmbH & Co. KG, Düren, Germany), then subsequently purified on Zymo-Spin IC to remove melanin (Zymo Research, Irvine, CA, USA). DNA concentrations were quantified by Qubit (Thermo Fisher Scientific, Waltham, MA, USA).

### 2.3. Libraries Preparation and Sequencing

Targeted-sequencing libraries were prepared using the Ion Ampliseq Comprehensive Cancer Panel (400 genes, 1.75 megabases (Mb), Thermo Fisher Scientific, Waltham, MA, USA) from 200 ng of DNA. The full list of interrogated genes can be downloaded here: https://assets.thermofisher.com/TFS-Assets/CSD/Reference-Materials/ion-ampliseq-cancer-panel-gene-list.pdf; Last accessed on 30 June 2021. Libraries were 100 bp paired-end multiplex sequenced on the Illumina HiSeq 2500 (Illumina Inc., San Diego, CA, USA).

### 2.4. Sequencing Analyses and Mutation Calling

Sequencing quality was assessed by FastQC (http://www.bioinformatics.babraham.ac.uk/projects/fastqc/; Last accessed on 30 June 2021). Reads were aligned to the human genome (hg19) with Bowtie2 2.1.0 (http://bowtie-bio.sourceforge.net/bowtie2/index.shtml; Last accessed 30 June 2021) [26]. PCR duplicates were removed using Picards Tool MarkDuplicates v1.97 (https://broadinstitute.github.io/picard/; Last accessed on 30 June 2021). Variant calling for SNP and indels were performed using HaplotypeCaller [27] (https://gatk.broadinstitute.org/hc/en-us; Last accessed on 30 June 2021). Variants were annotated using ANNOVAR [28] (http://annovar.openbioinformatics.org/en/latest/; Last accessed on 30 June 2021) with the databases ensGene, avsnp147 [29], cosmic81 (https://cancer.sanger.ac.uk/cosmic; Last accessed on 30 June 2021) [30], and popfreq all. Variants were filtered sequentially by (i) removing non-exonic variants; (ii) synonymous variants; (iii) variants with population frequency higher than 1% (ANNOVAR popfreq_all > 0.01); and (iv) variants covered with less than five reads of position depth (DP) and/or less than three reads of allele depth (AD) and/or a frequency (AD/DP) of less than 10%. Variants from genes suspected to be tumor suppressor genes (*BAP1*, *NF1*, *CDKN2A*, *ARID2*, *TET2*, *RB1*, or *PTEN*) were classified following MutationTaster scores and only variants predicted as non-polymorphic were kept (http://www.mutationtaster.org/; Last accessed on 30 June 2021) [31]. Mutations were described as recurrent in the COSMICv81 database if ≥3 occurrences in eye samples and/or ≥10 in skin and/or ≥15 in the whole database.

### 2.5. Statistics

Associations between qualitative features were tested using the Fisher’s exact test, while association between age and mutational status was tested using the Mann–Whitney test. Analyses were carried out with the R software v 4.0.3 (http://www.R-project.org/; Last accessed on 30 June 2021) [32] and a two-tailed *p* < 0.05 was deemed significant.

## 3. Results

### 3.1. Overview

Samples from 47 ConjMel patients treated between May 2004 and June 2016 at the Institut Curie were sequenced with a commercial panel of 400 genes (1.75 Mb) including genes implicated in melanoma oncogenesis. Clinical histories and pathology findings are described in Table 1 and Appendix A. We did not find a family history of cancers, in particular skin and conjunctival melanoma. None of our cases presented clinically detectable Ota nevus or uveal melanoma. Median follow-up was of 59.8 months. Median age was 69 years (range 44–93 years) with an equilibrated sex-ratio (23 men for 24 women). Eight tumors (17%) were entirely located in non-sunlight-exposed zones (i.e., behind the eyelid). They originated from primary acquired melanocytosis (PAM) in 24 cases (Figure 1; 51%), from nevi in eight cases (17%), and de novo in 15 cases (32%). The patients had a median of two surgeries (not including biopsies or brachytherapy, range 1–8, Table 1), 6/47 had an exenteration (13%), 8/47 (17%) suffered metastatic recurrences, and 17/47 (36%) had passed away at the last follow up. Four patients (9%) received local mitomycin C and 12 (26%) received radiotherapy before sampling.

### 3.2. Mutated Genes

Sequencing resulted in a median depth of 565× (range 190–3200×) and a median 20× coverage of 97.7% (range 69–99%). After filtering out low depth variants and known polymorphisms, a median of 786 variants per case was observed, most of them probably being germline variants as tumor sequencing was not matched with germline (range 544–9016). Two sun-exposed tumors presented outlier mutation burdens with a 2.3 and 11.4-fold increase of variants, suggesting a different mutational process, but subsequent whole-exome sequencing showed an ultraviolet-related signature. Variant analysis then focused on (i) tumor suppressor genes and known melanoma hotspots; (ii) mutations known to be recurrent in the COSMIC pan-cancer database; and (iii) other oncogenic mutations, namely mutations of *CBL*, *CTNNB1*, *RUNX1*, *SMARCA4*, and *TET2*. More than half of the cases presented mutually exclusive activating mutations in known melanoma oncogenes such as *BRAF* p.G464/G466/G469/V600/K601 in 16 cases (34%), *NRAS* p.G12/G13/Q61 in five cases (11%), *HRAS* p.G12/G13 in two cases, and *KIT* p.L576P in two cases (Figure 2 and Appendix A). Notably, *BRAF* p.G466E and p.G469R co-occurred with less potent *HRAS* p.G12S in case #37. *NF1* mutations were present in 17 cases including four *BRAF*-mutated samples. Typical uveal melanoma-related mutations were found in seven cases (15%): one case with *GNAQ* p.R183Q, one with *GNA11* p.R183C, two with *SF3B1* p.R625C/H, and three with *BAP1* missense/nonsense supposedly somatic mutations. No tumor presented concomitant variants in *GNAQ/GNA11* and in *SF3B1* or *BAP1*.

*CTNNB1* was found mutated in eight cases (17%). All eight mutations were located between codons 41 to 51 in the β-transducin repeat-containing protein (β-TrCP) binding motif. *CBL* was found to be mutated in four cases (9%). All mutations were missense variants located between codons 390 to 419 in the RING finger domain. Hotspot mutations of *TP53* were found in four cases (9%). Several mutations in the tumor suppressor genes were predicted to be deleterious including *BAP1*, *NF1*, *CDKN2A*, *ARID2*, *TET2*, *RB1*, *PTEN*, *XPC*, *PBRM1*, or *ATR*, but the definitive status of these genes could not be determined in the absence of germline sequencing. Finally, single occurrences of targetable oncogenic mutations were observed in *ERBB2* p.S310F, *IDH1* p.R132C, *MET* p.T1010I, and *MAP2K1* p.P124S.

### 3.3. Correlation with Clinical and Pathological Findings

The eight tumors originating from conjunctival nevi were enriched in CTNNB1 mutations (five out of eight; 63% vs. three out of 39 non-nevi MelConj (8%); Fisher’s exact test *p* = 0.001). Interestingly, 5/8 CTNNB1-mutated cases were devoid of BRAF, RAS, NF1, or KIT mutations. No specific genetic characteristic was found in PAM-originating ConjMel. Sunlight-exposed ConjMel was more often associated with BRAF, HRAS, NRAS, and CTNNB1 mutations than unexposed ConjMel (66.7% vs. 12.5%, respectively; Fisher’s exact test *p* = 0.007). Among the eight non-sunlight-exposed ConjMel, samples from two patients of more than 80 years bore the only two KIT activating mutations found in the series (2/8 unexposed cases vs. 0/39 exposed cases; Fisher’s exact test *p* = 0.026). These KIT mutations co-occurred with the only two SF3B1 oncogenic mutations found in the series. BRAF-mutated and CDKN2A-mutated ConjMel occurred in younger patients with median ages of 64 years vs. 73 years (Figure 3; Mann–Whitney test *p* = 0.070) and 60 years vs. 69 years (*p* = 0.18), respectively, while NF1-mutated cases only tended to be older (73 years vs. 69 years; *p* = 0.55). The eight cases with metastatic recurrences were RASm (three out of the six RASm cases), BRAFm (three out of 16), and RAS/BRAFwt (two out of 25). No link was found between genetic characteristics and treatment including prior exposures to mitomycin C treatment and radiotherapy.

## 4. Discussion

Accounting for only ~5% of ocular melanomas, the rarity and smallness of ConjMel have impeded large genetic studies. Our study describes the mutational analysis of 400 genes in one of the largest cohorts ever published. In our series, ConjMel shared features with chronically sun-damaged CM including older age at diagnosis and high incidences of *NRAS*, *KIT*, *BRAF* non-V600, and *NF1* mutations. These observations provide further evidence for a potential role of chronic, lifelong sunlight exposure in conjunctival melanomagenesis [12]. Indeed, UV-driven DNA damage is predominant in mucosal melanomas of conjunctival origin [16,17]. On the other hand, ConjMel also presented similarities with mucosal melanomas as they have been reported to display a high number of indels and copy number alterations [19]. Taken together, our data support that ConjMel is a biologically distinct, heterogeneous group of melanomas presenting a mixed phenotype with features of mucosal melanomas associated with genetic scars of chronic UV exposure similar to iris melanomas, a subset of uveal melanomas displaying a UV-induced, high mutation burden [33,34]. Similar to CM, such pathogenesis results in genetically defined ConjMel subsets in younger patients affected by *BRAF*- or *CDKN2A*-mutated ConjMel, while older patients carried *NF1*-mutated ConjMel, although the age difference was found to not be statistically significant [6]. However, the absence of germline sequencing in our study impedes definitive conclusions on the origin of inactivating variants in tumor suppressive genes such as *CDKN2A* or *NF1*. Recurrent mutations demonstrated a prominence of *MAPK* pathway activation, primarily through *NRAS* and *BRAF* activation, confirming previous reports [24,35]. ConjMel cases with *RAS* mutations may be of poor prognosis, but small sample size prevented us from drawing definitive conclusions. However, such association has also been reported in skin melanomas with *NRAS* mutation being identified as an independent predictor of disease progression [36]. We identified the same four subtypes as in CM, defined by *BRAF* (in 32% of cases), *RAS* (13% including the first report of *NRAS* p.G12R and *HRAS* activating mutations in ConjMel), NF1 mutations (22%, without concomitant *BRAF/RAS* mutation), or by the absence of these alterations (triple wild-type; 33%). This observation further solidifies the close relationship between CM and ConjMel. Overall, *BRAF* mutation incidence was at the intermediary level compared with other melanomas, ranging from ~50% in CM, and ~10% in mucosal to 0% in uveal melanomas [24,35]. As in CM, *BRAF* mutations were observed in younger ConjMel patients, confirming observations from Larsen and colleagues [4,35] including low activity *BRAF*nonV600E mutations (p.G464E, p.G466E, pS467L, and p.G469R). Similarly, we observed a subset of *NF1*-mutated ConjMel, mutually exclusive with other MAPK-activating mutations, in similar proportions than in CM [3,4].

The triple-wild-type cohort appeared to be heterogeneous with some cases carrying *CTNNB1* mutations, while others presented both *SF3B1/KIT* mutations. *CTNNB1*, coding for β-catenin, was found mutated in 15% of ConjMel cases, while they were exceptional in CM and mucosal melanomas (~5%) [4]. *CTNNB1* mutations were mutually exclusive from the other oncogenic mutations in most cases and associated with pre-existing nevi, suggesting a specific melanomagenesis process. However, the study of the origin of these cases was limited by the absence of reported analysis of pre-existing lesions in clinical files from two of our 47 patients. Of interest, *CTNNB1* mutations occur in an unusual type of melanocytic nevus, termed “deep penetrating-nevus” (DPN) occurring in the skin, the conjunctiva and in extremely rare melanomas derived from such DPN [37,38]. Among the eight *CTNNB1*-mutated tumors, clear cut evolution from DPN could not be confirmed (or ruled out) histopathologically with complete certainty, probably because of the evolution of the invasive component. *CTNNB1* mutations impede β-catenin degradation, provoke nuclear translocation, and promote transcription of target genes. Two samples presented T41A, a known oncogenic mutation present in half of desmoid tumors (1252/2422 cases in COSMIC). Other ConjMel presented missense variants in the β-TrCP binding motif between codons 48 to 51. Although less frequent than mutations in codons 32 to 45, variants in codons 48 to 51 have been identified in several other rare tumor types such as anaplastic thyroid carcinomas [39], sinonasal NK-T cell lymphomas [40], gastrointestinal carcinoid tumors [41], and gastric carcinomas intestinal type [42], supporting their oncogenic role in ConjMel. Two sun-protected ConjMel tumors presented concomitant *KIT* and *SF3B1* activating mutations. While *KIT* mutations have been observed at a similar overall rate in CM [3,4], they did not overlap with *SF3B1* mutations in these tumors. Rare activating *KIT* mutations and *SF3B1* mutations have been previously reported in ConjMel cases [23,24,25,43,44]. Although not emphasized in previous publications, some mucosal melanomas may carry concomitant *KIT/SF3B1* mutations [45,46,47,48]. Interestingly, all these cases were sun-protected mucosal melanomas with high incidence of these *KIT/SF3B1* co-mutations in ~20–40% of anorectal melanomas and ~33% of vulvar or genitourinary melanomas. *KIT* oncogenic mutations activate the downstream PI3K and MAPK pathways, while *SF3B1* mutations induce an alternative mRNA splicing. These two mutations may cooperate to induce the cancer phenotype as observed in around 20–25% of uveal melanomas [49]. Our data suggest that mutant *KIT/SF3B1* co-occurrence may be a specific feature of a subset of mucosal melanomas including ConjMel, anorectal, and genitourinary melanomas. Some cases presented with mutations usually linked to uveal melanomas (*GNAQ*, *GNA11*, *SF3B1*, *BAP1*), but these mutations did not co-occur, suggesting a different tumorigenesis process than in UM. The tumorigenesis of these tumors may be similar to that of rare skin melanomas exhibiting these UM-like mutations. Other rare recurrent, targetable, oncogenic mutations were found in our cohort including *IDH1* R132C, *ERBB2* S310F, *MET* T1010I, and *MAP2K1* P124S [43,50,51,52,53]. CBL missense variants were present in 6% of cases. CBL codes for an E3-ubiquitin ligase that promotes the ubiquitination of signaling proteins through the catalytic activity of its RING finger domain. Oncogenic mutations in codons 390, 417, and 419, as found in our cohort, belong to the RING domain and have been previously reported in myelodysplasia [54], acute myeloid leukemia [55], myeloproliferative syndromes [56], and chronic myelomonocytic leukemia [57].

## 5. Conclusions

Taken together, our data support the fact that ConjMel is a biologically distinct group of melanomas with features reflecting their mucosal origins as well as their chronic exposure to sunlight. Furthermore, this work promotes the emergence of personalized therapies for ConjMel patients. High mutation burden urges the use of checkpoint inhibitors in the metastatic as well as in the adjuvant settings while *BRAF* and *KIT* activating mutations qualify these patients to BRAF/MEK and KIT inhibitions. Ultimately, the discovery of new oncogenic, targetable mutations in *IDH1*, *ERBB2*, *MET*, and *MAP2K1* opens new therapeutic avenues as new pharmacological inhibitors are developed.

## Figures and Tables

**Figure 1 cancers-13-03836-f001:**
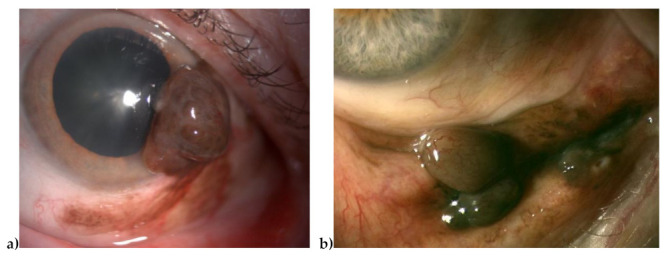
Two pictures of ConjMel originating at the limbus, from primary acquired melanocytosis ((**a**) case #4), and inferior fornix ((**b**) case #46).

**Figure 2 cancers-13-03836-f002:**
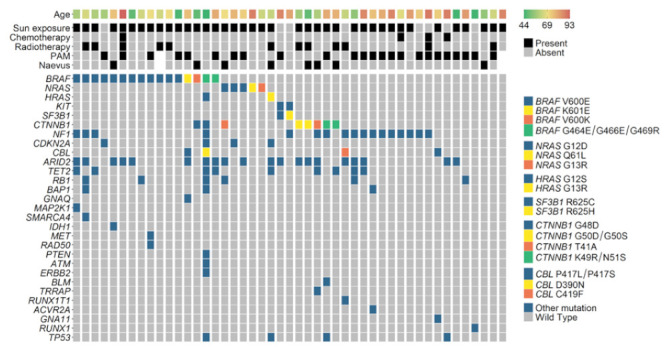
OncoPrint of mutations in melanoma-associated genes. PAM: primary-acquired melanocytosis.

**Figure 3 cancers-13-03836-f003:**
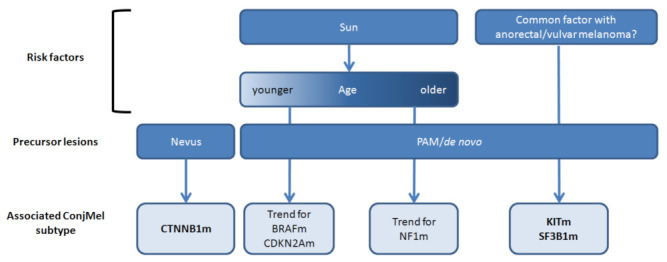
Proposal of molecular classification according to potential etiological factors.

**Table 1 cancers-13-03836-t001:** Summary of clinical features.

Characteristics	Values
Total number of patients	47
Median age at sampling (years)	69 (range 44–93)
Gender	
Female	24/47 (51%)
Male	23/47 (49%)
Tumor location	
Sun-exposed	39/47 (83%)
Non-sun-exposed	8/47 (17%)
Premalignant lesion	
PAM	24/47 (51%)
Nevus	8/47 (17%)
De novo	13/47 (28%)
Unknown	2/47 (4%)
Tumor stage	
T1	24/47 (51%)
T2	12/47 (26%)
T3	8/47 (17%)
unknown	3/47 (6%)
Histology	
Epithelioid	21/47 (45%)
Spindle cells	7/47 (15%)
Mixed	13/47 (28%)
Unknown	6/47 (13%)
Mitotic index	
Low (<11 figures/10 fields)	18/47 (38%)
Intermediate (11–22 figures/10 fields)	8/47 (17%)
High (>22 figures/10 fields)	14/47 (30%)
Treatment	
Median number of surgeries (range)	2 (1–8)
Number of exenterations	6/47 (13%)
Adjuvant treatment before sampling	
adjuvant mitomycine C	4/47 (9%)
adjuvant radiotherapy	12/47 (26%)
Recurrences	
Number of metastatic recurrences	8/47 (17%)

## Data Availability

Sequencing data available on demand from the corresponding author.

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
