# Peer review of "Definition of Biologically Distinct Groups of Conjunctival Melanomas According to Etiological Factors and Implications for Precision Medicine"

_cancers, 2021, doi:10.3390/cancers13153836_

Round 1
Reviewer 1 Report
The authors have responded to all my previous concerns. My only issue pertains to variant calling and predictions of LOH. The authors state that they are unable to do this: "However we cannot determine the copy number of these 400 genes in the absence of a genomic backbone, thus impeding any conclusion on LOH or tumor cell content...". However, the VAF is sufficient to determine if there has been LOH in tumors. It can only be > 0.5 if there was loss of the other allele - the fact that it is often less than 1 is due to contaminating normal cells or tumor heterogeneity. When this reviewer investigated the genes with high VAFs in the revised Supplementary Data it was evident that there are were number of novel and plausible tumor suppressors that harbor mutations in several tumors that should be mentioned and that would significantly add to the impact of the manuscript. These included XPC, PBRM1, BAP1 and ATR. I am wondering why the authors didn't mention these (BAP1 is shown in Figure 2) or look at these or other plausible candidates further.
Author Response
The authors have responded to all my previous concerns. My only issue pertains to variant calling and predictions of LOH. The authors state that they are unable to do this: "However we cannot determine the copy number of these 400 genes in the absence of a genomic backbone, thus impeding any conclusion on LOH or tumor cell content...". However, the VAF is sufficient to determine if there has been LOH in tumors. It can only be > 0.5 if there was loss of the other allele - the fact that it is often less than 1 is due to contaminating normal cells or tumor heterogeneity. When this reviewer investigated the genes with high VAFs in the revised Supplementary Data it was evident that there are were number of novel and plausible tumor suppressors that harbor mutations in several tumors that should be mentioned and that would significantly add to the impact of the manuscript. These included XPC, PBRM1, BAP1 and ATR. I am wondering why the authors didn't mention these (BAP1 is shown in Figure 2) or look at these or other plausible candidates further.
We thank reviewer #1 for his/her remarks.
The 0.5 threshold is appropriate if (i) we are analyzing a pure tumor sample and (ii) the locus is diploid. If the tumor cell content is lower, the threshold must be reduced. If the locus is not diploid, the threshold must be changed. The absence of copy number assessment thus impedes any conclusion on LOH.
We did mention several TSG and are now including those cited by reviewer #1:
“Several mutations in tumor suppressor genes were predicted to be deleterious including BAP1, NF1, CDKN2A, ARID2, TET2, RB1, PTEN, XPC, PBRM1 or ATR”. However, the somatic origin of these variants in TSGs cannot be affirmed without germline DNA as stated in the results section: “Several mutations in tumor suppressor genes were predicted to be deleterious […] but the definitive status of these genes could not be determined in the absence of germline sequencing.” and in the discussion “However, the absence of germline sequencing in our study impedes definitive conclusions on the origin of inactivating variants in tumor suppressive genes such as CDKN2A or NF1.”
Reviewer 2 Report
We thank the authors for their revised manuscript, which describes gene mutations through NGS in a cohort of conjunctival melanoma (conjmel) with the aim of determining disease-specific mutations in comparison to other melanoma subtypes.
The manuscript is well-written and easy to follow.
Major comment:
The differentiation between sun-exposed vs non-sun exposed is ambiguous. It does not make clear distinction between fully, mostly and minimally exposed conjmel. What is meant by ‘ at least a part is exposed’?, how many millimetres were considered exposed? What if the tumor bulk is non-exposed but there is associated partially exposed PAM or mere flat pigmentation? The standard description of conjmel is the location being ‘bulbar’ and ‘non-bulbar (forniceal, tarsal and caruncular). Therefore, the whole manuscript should be written with such description rather than the unclear distinction used.
Minor comments:
- In abstract: “Targetable mutations were 69 observed in ERBB2, IDH1, MET and MAP2K1, opening new therapeutic avenues”; please provide percentages detected
- Ideally, as the manuscript is related to conjmel, the Introduction section should commence with conjmel then compare known literature to other melanomas
- ConjMel origin line 109: please state the de novo percentage
- The cause for increasing incidence of melanoma line 111 must be stated
- The sentence ‘median of 2 surgeries” is not clear as to what kind of surgeries, but this is clear in the supplementary table, so the authors should write “refer to supplementary table…”
- In supplementary table 1: what is last date news?? Is it last review date? Is it date of end of study? Please clarify
- The authors mention several mutations but it is not clear how the 2 conjmel of unknown origin were accounted for in this statistical analysis? Perhaps just Please explain this briefly in the discussion that it was not possible or if they have been excluded from the analyses of such and such…etc
- Figure 3 should be omitted. It is not adding anything to the txt, especially that most of the results included in the table were not statistically significant
- Line 264: “Our study is an extensive description…”, please explain that this was not exhaustive
- Line 279: sentence ” NF1-mutated conjmel…” the authors state that there is a trend to be associated with older age, whereas their statistics show non significance (p=0.55). The text must state they are not statistically significant and delete ‘trend’
- In the discussion, several variants and mutations are mentioned that need to be stated in the results section first with their percentages such as β-TrCP then discuss their significance, if any, in the discussion section.
- In discussion line 270: the sentence “On the other hand….”. Please move this sentence be moved into the introduction because no work on indels has been performed in the current manuscript.
Author Response
We thank the authors for their revised manuscript, which describes gene mutations through NGS in a cohort of conjunctival melanoma (conjmel) with the aim of determining disease-specific mutations in comparison to other melanoma subtypes.
The manuscript is well-written and easy to follow.
We thank reviewer #2 for his/her remarks.
Major comment:
The differentiation between sun-exposed vs non-sun exposed is ambiguous. It does not make clear distinction between fully, mostly and minimally exposed conjmel. What is meant by ‘ at least a part is exposed’?, how many millimetres were considered exposed?
We did a binary analysis between non-exposed and any level of sun exposure. Sunlight exposure was defined based on careful review of the location of the tumors (pictures and clinical schemes). To be considered “sun-protected”, the tumors had to be fully protected by the eyelids. Any exposed part of the tumor, whatever the size, would make this tumor be considered “sun-exposed”.
What if the tumor bulk is non-exposed but there is associated partially exposed PAM or mere flat pigmentation?
If the tumor bulk was non-exposed, then the tumor was considered non-exposed.
The standard description of conjmel is the location being ‘bulbar’ and ‘non-bulbar (forniceal, tarsal and caruncular). Therefore, the whole manuscript should be written with such description rather than the unclear distinction used.
We understand this point and it would be of interest if we were exploring prognostic factors, including the AJCC staging system. However, the goal of our work is to document mutations in these tumors and to compare with potential etiological factors, not with prognostic factors. For this reason, the distinction between bulbar and non-bulbar is irrelevant as this distinction is not associated with sun exposure. For instance non-bulbar caruncle is exposed but not the fornices.
Minor comments:
- In abstract: “Targetable mutations were 69 observed in ERBB2, IDH1, MET and MAP2K1, opening new therapeutic avenues”; please provide percentages detected
Modified:
“Targetable mutations were observed in ERBB2, IDH1, MET and MAP2K1 (one occurrence each).”
- Ideally, as the manuscript is related to conjmel, the Introduction section should commence with conjmel then compare known literature to other melanomas
We thank the reviewer for this suggestion, but we decided to keep the context in our order.
- ConjMel origin line 109: please state the de novo percentage
Modified:
Conjunctival melanoma (ConjMel) arises from conjunctiva, the ocular external mucosa, mostly within areas of primary acquired melanocytosis (PAM) (~65% of cases), in conjunctival nevi (~15% of cases) or de novo in ~20% of cases [10-12].
- The cause for increasing incidence of melanoma line 111 must be stated
Modified:
“[…] its incidence has been increasing over the last decades with a suspected relationship to sunlight exposure [13,14].”
- The sentence ‘median of 2 surgeries” is not clear as to what kind of surgeries, but this is clear in the supplementary table, so the authors should write “refer to supplementary table…”
The type of surgery is not provided in Supp Table 1, but the number of exenterations is provided in Table 1 (now cited in the results section). Because the goal of this article is to explore the mutation profile of ConjMel, which is not influenced by surgery, we did not comment the type of surgeries.
- In supplementary table 1: what is last date news?? Is it last review date? Is it date of end of study? Please clarify
It corresponds to the last time we saw the patient (if alive) or time of death. Modified to “date of last follow-up”.
- The authors mention several mutations but it is not clear how the 2 conjmel of unknown origin were accounted for in this statistical analysis? Perhaps just Please explain this briefly in the discussion that it was not possible or if they have been excluded from the analyses of such and such…etc
We did not remove the 2 conjmel of unknown origin for the statistical analysis. We modified the discussion section to comment this point.
“However, the study of the origin of these cases was limited by the absence of reported analysis of pre-existing lesions in clinical files from two of our 47 patients.”
- Figure 3 should be omitted. It is not adding anything to the txt, especially that most of the results included in the table were not statistically significant
We kept this summary Figure 3, as it may be helpful for the readers. However, we modified according to the important referee’s remark, to reflect the non-significant trend associated with age and we changed the title to “Proposal of molecular classification according to potential etiological factors”.
- Line 264: “Our study is an extensive description…”, please explain that this was not exhaustive
Modified to:
“Our study describes the mutational analysis of 400 genes in one of the largest cohorts ever published.”
- Line 279: sentence ” NF1-mutated conjmel…” the authors state that there is a trend to be associated with older age, whereas their statistics show non significance (p=0.55). The text must state they are not statistically significant and delete ‘trend’
Modified to:
“Similar to CM, such pathogenesis results in genetically defined ConjMel subsets in younger patients affected by BRAF- or CDKN2A-mutated ConjMel, while older patients carried NF1-mutated ConjMel, although the age difference was found not statistically significant [6].”
- In the discussion, several variants and mutations are mentioned that need to be stated in the results section first with their percentages such as β-TrCP then discuss their significance, if any, in the discussion section.
β-TrCP is a domain of the β-catenin protein. All numbers of mentioned mutations are given in the results section, including the β-TrCP CTNNB1 mutations.
Quote from the results section:
“CTNNB1 was found mutated in eight cases (17%). All eight mutations were located between codons 41 to 51 in the β-transducin repeat-containing protein (β-TrCP) binding motif.”
- In discussion line 270: the sentence “On the other hand….”. Please move this sentence be moved into the introduction because no work on indels has been performed in the current manuscript.
Introduction modified to:
“Recent reports showed that ConjMel present an UV-associated mutational signature together with a high number of indels and copy number alterations [17-19].”
This manuscript is a resubmission of an earlier submission. The following is a list of the peer review reports and author responses from that submission.
Round 1
Reviewer 1 Report
The authors describe the results of panel-based sequencing of 47 conjunctival melanomas from FFPE specimens – 400 genes were interrogated. The authors detected CTNNB1 mutations in nevi and additional mutations in BRAF/CDKN2A, NF1 and KIT/SF3B1. The study was technically well performed, and the rests are of interest given that CM is so rare. Unfortunately matched normal DNA was not available so that somatic mutations could not be definitively identified and differentiated from germline alterations. However, the manuscript could benefit from additional details listed below:
Please provide the 400 genes that were interrogated as supplementary data.
In Supplementary Table 3 the authors should provide the frequency of mutation reads in tumors. For critical tumor drivers discussed in the text, could they also discuss whether these were present at frequencies suggesting gain of function alterations, germline passengers (with frequencies of 50%) or have undergone LOH (frequencies > 50% in tumors). Along these lines could the authors please estimate the proportion of tumor in each of the samples. This will also help with interpretation of variant allele frequencies.
The authors indicate that the high incidence of certain mutations provide evidence for a role for sun exposure as a trigger in CM. Can they determine if tumors have a UV signature to support this hypothesis.
The authors indicate that for CM, tumors are frequently multifocal suggesting germline susceptibility. Can the authors comment on this – and point out that they are in able to investigate this thoroughly since they were not able to analyze matched normal tissue from patients with tumors. Hence, they may have missed detecting critical germline alterations.
Supplementary Table 2 seems to just be a dump from mpileup. Moreover, why do the authors feel it is necessary to provide variants where DP or AD is 0?
Ensure that sample IDs are consistent in supplementary tables to permit easy cross-referencing.
As indicated above add variant allele frequencies to supplementary Table 3 and indicate if the change is likely to be heterozygous or homozygous (or hemizygous) in a separate column. In this table also provide output from ANNOVAR, including cDNA changes, pathogenicity, whether this variant was identified previously in tumors and its COSMIC ID, SNP ID if available etc.
Page 6: Delete “The text continues here.”
Reviewer 2 Report
cancers-1249488
The authors report on the sequencing of 400 cancer genes in 47 primary ConjMel using the commercially available Ion Ampliseq Comprehensive Cancer Panel from Thermo Fisher. The study confirms earlier results reported bij Scholz et al. (your ref 23). Several associations between mutations and etiological factors were investigated. They describe that BRAF- and CDKN2A-mutated ConjMel affect younger patients while NF1-mutated tumors tend to develop in the elderly. CTNNB1 mutations were more frequent in nevi-derived ConjMel indicating that the Wnt pathway is pivotal in tumorigenesis in this type of melanoma. Concomitant KIT/SF3B1 mutations in BRAF/RAS-wild type, sun-protected tumors are reported. The study is straightforward, well described, well performed and adds to the current understanding of differences in melanoma subtypes. Major and minor comments can be made.
Major comments
1 CTNNB1, coding for β-catenin, was found mutated in 15% of ConjMel cases. The tumors originating from conjunctival nevi were enriched in CTNNB1 mutations (5 out of 8; 63% versus 3 out of 39 non-nevi MelConj (8%); Fisher’s exact test 238 p=0.001).
It is well known that CTNNB1 mutations are specific for deep penetrating nevi (DPN). DPN have recently been described to occur frequently in the conjunctiva (PMID: 31558492) and CTNNB1 mutations can be demonstrated n such cases (PMID: 33005620). Please comment on the exact type of nevus associated with the reported CTNNB1 positive ConjMel described here. In general, DPN in the skin are rarely associated with melanoma.
2 Typical uveal melanoma-related mutations were found in seven cases (15%): one case with GNAQ p.R183Q, one with GNA11 p.R183C, two with SF3B1 p.R625C/H and three with BAP1 missense/non- sense supposedly somatic mutations. No tumor presented concomitant variants in GNAQ/GNA11 and in SF3B1 or BAP1.
GNAQ/11 mutations are associated with blue nevi in the skin and periorbital area. Please comment on the supposed origin of the cases described here.
BAP1 mutations have been well described in uveal melanoma to co-occur with monosomy 3. Immunohistochemistry for BAP1 protein in such cases is negative. Was BAP1 immunohistochemistry performed? May these BAP1 mutations be viewed as passenger mutations or do the authors expect these to be important for tumorigenesis in ConjMel? Please indicate that the possibility of extrascleral extension of anterior located uveal melanoma through the water veins was excluded.
3 8/47 188 (17%) suffered metastatic recurrences, median follow-up was of 59.8 months
Please provide information on the associations of the mutations described to prognosis is possible.
4 Rare activating KIT mutations have been previously reported in ConjMel cases but SF3B1 was never sequenced in these series SF3B1
Earlier reports on SF3B1 mutations in ConjMel have been overlooked. Please provide the proper references. Such as reference 23 in the current manuscript and PMID: 33156594 and PMID: 34071371
Minor comments
1 Taken together, our data support that ConjMel is a biologically distinct, heterogeneous group of melanomas presenting a mixed phenotype with features of mucosal melanomas associated with genetic scars of chronic UV exposure similarly to iris melanomas, a subset of uveal melanomas displaying a UV-induced, high mutation burden
A reference to the largest cohort of iris melanomas investigated for recurrent mutations to date has been overlooked (PMID: 29371009).
2 Other new recurrent, targetable, oncogenic mutations were found in our cohort including IDH1 R132C, 316 ERBB2 S310F, MET T1010I and MAP2K1 P124S.
References to earlier reports on these mutations in ConjMel have been overlooked (your ref 23, PMID: 33156594). Please check the literature more stringent when claiming new findings.
3 Figure 1. Two pictures of ConjMel originating at the limbus, from primary acquired melanocytosis (left) and inferior fornix (right). Figure 2. OncoPrint of genomic alterations in melanoma-associated genes. PAM: primary-acquired melanocytosis
Please provide unique identifiers in table 2 and provide the unique identifiers for the cases presented in figure 1.
Reviewer 3 Report
We thank the authors for their work. The study examines gene mutations through NGS in a cohort of conjunctival melanoma (conjmel) with the aim of determining mutations specific to aetiological factors
The methods are described reasonably well, and the cohort size is one of the larger in literature to examine conjmel genetics, so we congratulate the authors on that.
However, there are some issues with the work:
- In the simple summary: the sentence “The relationships between potential etiological factors such as ultraviolet exposure and ConjMel mutational landscape have not been precisely described”, line 48, is misleading because there is a myriad of articles describing ConJMel mutations albeit more work is needed to understand the UV role on a large scale. It would be better to reword this by saying the UV role in ConjMel is still debatable because of the lack of large cohort studies.
- In simple summary: “We further identify concomitant KIT/SF3B1 mutations in BRAF/RAS-wild type, sun-protected tumors…..”, If the authors did not examine these mutations in sites other than the conjunctiva then no need to say “and show that the same profile can be found in genital and anorectal melanomas, unveiling a distinct, mucosal-specific, tumorigenic pathway”. This part of the sentence should be deleted or reworded.
- several English language errors such as the word enriched on line 65, it should read “rich”, depending “of line 97 should read “on”,…etc, line 334 exposition should read ‘exposure’
- In abstract: “ The eight tumors arising from nevi were enriched in CTNNB1 mutations (63% 65 versus 8%; Fisher’s exact p-test = 0.001)”, what is the 8% referring to? What is being compared??
- In abstract: “The two KIT-mutated cases carried SF3B1 mutations and were …” must be referenced.
- In abstract: “Targetable mutations were 69 observed in ERBB2, IDH1, MET and MAP2K1, opening new therapeutic avenues”: percentages detected of these mutations must be stated because they may not be of value if they are of low prevalence.
- The Introduction section starts with a long paragraph on Uveal (UM) and Cutaneous Melanoma (CM) whereas the article is primarily on conjmel, therefore it would be more appropriate to commence the introduction by the conjmel section then how it compares to other melanoma subtypes.
- the Introduction section about the various melanoma subtypes states: “Altogether, these genetic alterations influence clinical decisions for disease management….”. It must be made clear that the management is in relation to metastatic and not primary disease. Same for checkpoint inhibitors
- ConjMel origin line 109 must state the den novo percentages
- The cause for increasing incidence of melanoma line 111 must be stated
- “ConjMel risk factors…” line 112 should not include UV exposure as a risk factor but could be mentioned as a ‘potential’ risk factor that is yet debatable because of lack of large studies on the subject.
- line 115”conjmel is frequently multifocal” is not true and the majority of cases are unifocal. If the authors have a reference that states the frequency of multifocality then they should add it to the text. In addition, recurrences are not frequent unless the primary tumor has not been treated adequately. The authors need to ascertain their statements with adequate references.
- Line 116 “Mitocmin C…”: The authors have not referred at all to the primary treatment of excision and what is available in the literature regarding adjuvant therapy. The treatment modalities need to be stated in order of commonness/effectiveness.
In addition, Treatments are not to avoid exenteration, but to prevent recurrences and mets..
- Line 120: Conjmel is not primarily sight threatening! This needs to be reworded.
- line 122: we do not agree with the authors that conjmel genetics has not been extensively explored because there is a myriad of genetic studies and all previously published genetic alterations in Conjmel should be stated in the introduction before mentioning UV signature. The authors should reword perhaps by stating that there is a paucity of large-cohort genetic CoM studies
- Line 132: how did the authors define sunlight exposed tumors?? What is their definition of ‘ at least a part is exposed’? How big is that part? What if the conjmel originated behind eye lid but later extended?? This is very misleading. The standard classification of conjmel including in the AJCC staging system is according to being ‘bulbar’ and ‘non-bulbar (forniceal, tarsal and caruncular). Therefore, the whole manuscript should be written with such description rather than the arbitrary distinction described. In addition, if genetic anomalies are described here, why haven’t the authors explored the relationship for such abnormalities to pathological descriptives, or even recurrences or metastasis??
- line 163: ‘non-exonic≥ variants”, what does this symbol ‘≥’ signify??
- Overview of results: There are lots of descriptives here than need to be addressed including table 1 and supplementary table 1
What is ‘yo’? if it is years then should be written ‘yrs’
What is equilibrated sex-ratio? Please be clear in descriptives
What is ‘median of 2 surgeries”? what kind of surgeries? Excision/ biopsy/plaque??? Please be clear and specific
36% had metastatic recurrences, what does that mean?? All patients with recurrence had metastatic disease?? Have they all died because of metastasis or other causes??
What power field has been used for mitotic index?
In supplementary table 1: what is last date news?? Please improve on English language
- In the results section of mutated genes:
The authors repeatedly say ‘probably’ and supposedly’ and ‘ but the definitive status of these genes….’, which undermine the validity of their results. Why haven’t validation tests been undertaken?
2 tumors had outlier mutations, the authors mention that they later showed UV signature, but they did not mention whether these 2 tumors were from sun exposed area or not
- figure 2: the legend mentions genomic alterations, but the text of the manuscript is about mutations, so can this be clarified in the figure legend??
- In section of correlation to clinical and pathological findings:
The authors mention CTNNB1 mutation in conjmel from naevus origin vs others. How were the 2 conjmel of unknown origins been accounted for in this statistical analysis?
What is meant by exogenous exposure? Please be specific
- Figure 3 should be omitted. It is not adding anything to the txt
- In discussion:
Line 258: “Our study is an extensive description…” this was not an exhaustive study and the authors must mention that
Line 261: ‘these observations…conjmel is linked to sun-tanning….’, this sentence is not justifiable because sun tanning was not examined in this series, so please delete this sentence or mention that this needs to be explored a s causative factor on a large scale.
Line 264: “On the other side,…’ the authors state that conjmel shares genetic similarities with mucosal melanomas because of high indels, whereas nowhere in the results indels were mentioned. The authors also state that there are probably high copy number alterations, if CNAs have not been examined then this phrase should be omitted. The manuscript seems to make suggestions throughout, which is misleading
Line 266: the sentence “ taken together,…” the word ‘similarly’ should be written ‘similar’. Again, same error in the following sentence
Line 270: sentence ” Similarly to CM, such pathogenesis,…..” the authors state that there is a trend of certain mutations to be associated with younger vs older age whereas their statistics do not show any significance. This is again a misleading sentence
The conjmel are described as a 4-group entity (mutations and triple wild type) and the wild type group is discussed in relation to CTNNB1 mutations found. It would have been easier to put all mutations in relation to clinical/pathological features in a table in the results then discuss in the discussion
Line 290: “ CTNNB1, coding…” the sentence needs a reference
In the discussion, several variants and mutations are mentioned without being described in the results and in very small numbers in the study cohort, e.g., T41A in 2 samples and β-TrCP in unknown number!! What is the significance of these alterations if they are in such minimal numbers? Why need to discuss which other bodily malignancies they’re present in?? It is expected in malignancy that there would be many outliers, which unlikely to be significant unless examined extensively or proven to be present in a significant number of samples.
Line 312: “These two mutations may cooperate…”. The sentence needs a reference
Line 315: “Other new recurrent……” what is the significance of such mutations? What was their frequency? Why weren’t they mentioned in the results? Again, CBL: was present in 6%, which is approx. 2-3 patients, is that significant in conjmel??
Main issues:
- The authors need to reclassify the tumors into bulbar and non bulbar then reanalyse
- It feels as if the authors are trying to convince the reader that all their results are important regardless of their frequency because were detected in other cancers. It would be better of the authors mention the detected alterations perhaps in a table with their frequencies then only discuss in the discussion sections the frequent ones in relation to clinical and pathological characteristic but only after rewriting the paper with conjmel separation into the standard bulbar and nonbulbar tumors.
- The title and conclusion focus on the importance of the findings for personalised therapies despite the fact that only local tumors were studied and not metastatic ones. So, unless the results are analysed in relation to metastasis or samples from metastatic tumors are used, the manuscript should not mention therapeutic targets unless there is justification for it or there are emergent therapies targeting genetic alterations/mutations in primary/recurrent local conjmel.
- The text in many areas does not give justice to previous CoM publications and is very shallow. The results and discussion sections need to be rewritten in view of the above comments.